# GeoPlant: Spatial Plant Species Prediction Dataset

**Lukas Picek**[1]**, Christophe Botella**[1]**, Maximilien Servajean**[2]**, César Leblanc**[1]**, Rémi Palard**[1]
**Théo Larcher**[1]**, Benjamin Deneu**[1]**, Diego Marcos**[1,3]**, Pierre Bonnet**[4]**, and Alexis Joly**[1]
[1] INRIA [2] Université Paul Valéry [3] Université de Montpellier [4] CIRAD, UMR AMAP

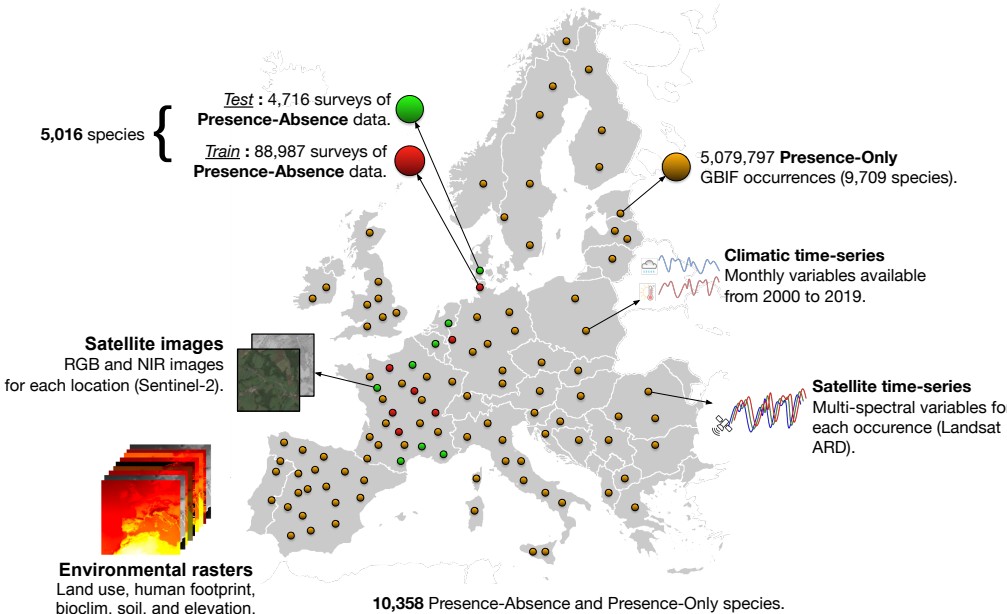

## Abstract

The difficulty of monitoring biodiversity at fine scales and over large areas limits ecological knowledge and conservation efforts. To fill this gap, Species Distribution Models (SDMs) predict species across space from spatially explicit features. Yet, they face the challenge of integrating the rich but heterogeneous data made available over the past decade, notably millions of opportunistic species observations and standardized surveys, as well as multimodal remote sensing data. In light of that, we have designed and developed a new European-scale dataset for SDMs at high spatial resolution (10–50m), including more than 10k species (i.e., most of the European flora). The dataset comprises 5M heterogeneous Presence-Only records and 90k exhaustive Presence-Absence survey records, all accompanied by diverse environmental rasters (e.g., elevation, human footprint, and soil) traditionally used in SDMs. In addition, it provides Sentinel-2 RGB and NIR satellite images with 10 m resolution, a 20-year time series of climatic variables, and satellite time series from the Landsat program. In addition to the data, we provide an openly accessible SDM benchmark (hosted on Kaggle), which has already attracted an active community and a set of strong baselines for single predictor/modality and multimodal approaches. All resources, e.g., the dataset, pre-trained models, and baseline methods (in the form of notebooks), are available on Kaggle, allowing one to start with our dataset literally with two mouse clicks.

38th Conference on Neural Information Processing Systems (NeurIPS 2024) Track on Datasets and Benchmarks.

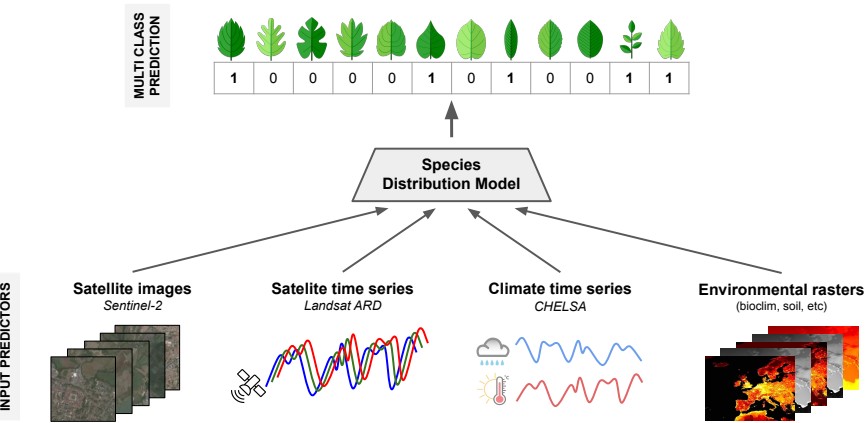

Figure 1: **Our view on Species Distribution Models (SDM)**. The SDM utilizes multimodal predictors (e.g., satellite, climate, and environmental data) for given GPS coordinates to predict multi-species compositions at that location.

# 1   Introduction

Global changes rapidly transform ecosystems, and their local impacts are context-dependent and hard to predict [17]. Monitoring species composition at high spatial resolution is crucial for understanding biodiversity responses and aiding decision-making, but has proven to be extremely challenging. Deep learning-based species distribution models (deep SDMs) [1, 7, 14] offer a promising venue by allowing to use high-resolution geographic predictors and remote sensing data to address sampling gaps [13, 22, 36] (see Figure 1). However, the heterogeneity, imbalance, bias and complexity of species observations and environmental data make model implementation challenging. Apart from that, standardized biodiversity data, i.e., exhaustive Presence-Absence (PA) surveys, are limited in coverage as they are time-consuming and costly to update and maintain. Even with a relatively large amount of PA data, it is difficult to model and map biological groups with large taxonomic diversity, such as plants, which have around 400k species to date, a vast majority of which are rare [21].

On the other hand, Presence-Only (PO) data from citizen-science platforms/initiatives (e.g., iNaturalist and Pl@ntNet), have emerged as valuable sources of large amounts of biodiversity data [3]. Even though those data have the potential to fill the distribution gap of the PA surveys, as they provide millions of geolocated records of tens of thousands of species annually, they are severely limited in that they do not indicate the absence of non-observed species and are heavily biased towards areas with a high density of observers [29]. Besides, the PO data represent a fraction of the species communities in regions with limited sampling and are biased toward common and/or appealing species [23]. As a result, incorporating PO data into SDMs risk introducing these biases [2, 42].

To allow a standardized use of available data and enable further research in ecological modeling, machine learning, and species distribution modeling, we have assembled a new European-scale dataset for Plant Species Prediction – **GeoPlant**. The dataset includes more than 5M heterogeneous PO records and 90k standardized PA surveys covering 10k+ species. All records are accompanied by (i) diverse environmental rasters (e.g., elevation, human footprint, and soil), (ii) Sentinel-2 RGB and NIR satellite images with 10 m resolution, (iii) a 20-year time series of climatic variables and (iv) satellite time series from the Landsat program. The GeoPlant dataset is the biggest dataset for species distribution modeling and the only dataset that includes satellite images and time series, climate time series, and environmental rasters. Besides, through the standardized PA data available in GeoPlant, model evaluation is made robust to the many biases of the PO data.

Following our successful long-term efforts in benchmarking SDM models [6, 32, 38, 39, 49], we also provide an openly accessible benchmark (hosted on Kaggle), which has already attracted an active community and established strong baselines for various single and multimodal approaches. With all needed resources (i.e., dataset, pre-trained models, and baseline methods) already publicly available, we create an ideal environment for benchmarking any new species distribution modeling approach.

## 2 Related Work

Species distribution models have relied for decades on geographic predictors at a spatial resolution of the order of a kilometer, such as bioclimatic [4], land cover [40] or human footprint variables [15]. At the same time, remote sensing data represent an unprecedented opportunity to provide high spatial resolution species distribution models with rich and globally consistent predictor variables describing the environment [31] and its temporal changes. However, its integration into species distribution models is recent and challenging [13, 22]. This data can complement the picture of the environmental landscape provided by the variables above at a coarser spatial scale. Yet, integrating variables at different spatial or temporal resolutions within deep learning architectures brings challenges.

**Open datasets and benchmarks** for species distribution modeling are still rare, and objective comparison of methods on existing datasets is limited [57]. The most cited benchmark [19], built in 2006, includes point location records for 226 anonymized species from six regions with accompanying predictor variables. Its key innovation was providing both PO and PA data to address spatial distribution biases in PA sites [48]. Our new dataset scales this approach up significantly, with about 100 times more occurrences and species, and includes more diverse predictors such as medium-resolution remote sensing data. **GeoPlant** allow evaluation of new SDMs, particularly those based on multimodal deep neural networks, and help identify fundamental factors determining species distribution. Another benchmark [10], published in 2021, uses forest inventory data across the western US to explore SDM extrapolation limits. This dataset covers 286,551 plots and focuses on 108 tree species with 19 bioclimatic predictors at a coarse spatial resolution (1 km). The newest dataset addition, SatBird [56], introduces a dataset for bird species distribution modeling based on satellite images, environmental data, USA-originating presence-absence data from the citizen science platform eBird and species range maps. In addition to dedicated benchmarks, more ecological studies are publishing their data openly, which allows their use for SDM evaluation [52, 59, 61]. However, these often suffer from small-scale and basic predictors based on tabular data. Recent work on deep SDMs [9, 14, 22] incorporates more complex predictors like images or time series but usually relies on PO data alone, introducing significant evaluation biases.

**The GeoLifeCLEF** is a long-term SDM evaluation campaign [5, 8, 33, 38, 39], organized by the authors, attracting considerable participation. GeoLifeCLEF aims to evaluate SDMs with unprecedented scope in species coverage, predictor multimodality, and spatial resolution (approximately 10 meters). Before 2023, the datasets were based solely on PO data with observational bias. In 2023, PA data were included as the main test set based on exhaustive survey data from the EVA [11]. The dataset presented here (i.e., GeoPlant) extends the 2023 dataset with additional PA data and new predictors, such as bioclimatic time series. It is unique in its scale (continental), spatial resolution (10m for the finer modality), and predictor diversity. From a MLperspective, it poses two major challenges: (i) multimodality, i.e., how to select and combine numerous heterogeneous information sources, and (ii) distribution shift, i.e., how to train effective models on PO data to predict PA.

**Available methods** suitable for species distribution modeling are usually divided into four groups:

- *Traditional statistical methods*, like generalized linear models [25, 27, 60], logistic regression [46], and Generalized Additive Models (GAMs) [62], form the backbone of SDM, with frameworks like Hierarchical Modeling of Species Communities (HMSC) [44] integrating additional data sources.

- *Traditional machine learning* include boosted regression trees, random forests, support vector machines, and neural networks, address complex species-environment relationships, with CNNs [12, 18, 20, 45] and provide advanced feature extraction from spatial environmental arrays.

- *Presence-only methods,* such as MaxEnt [47], estimate species observation probabilities based on environmental covariates and often use pseudo negatives for enhanced accuracy. A recent comparative study evaluated 13 different models, including both statistical and ML models [57]. However, since the evaluation was based on the dataset of Elith et al. [19], it did not allow the evaluation of more complex models, i.e., deep SDMs.

- *Deep SDMs* is a generic term for the new generation of SDMs using deep learning methods as a means of improving predictive performance and better understanding the contribution of complex factors such as spatial and temporal structures [7, 9, 14, 22]. Besides, it allows to work with various loss functions that tackle class imbalance or species pseudo-absences [63].

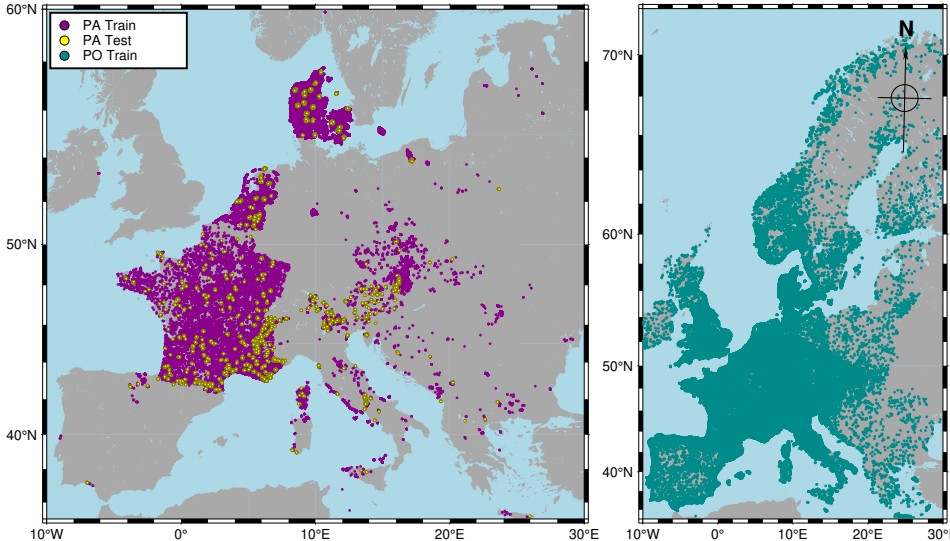

Figure 2: **Geo spatial scale of the dataset**. While the provided Presence-Only (PO) data spans all of habitable Europe, the Presence-Absence (PA) training and test sites are primarily from France, Denmark, Switzerland, and Czechia.

## 3  GeoPlant Dataset

The GeoPlant dataset comprises **Species Observation** (i.e., *Presence-Only* occurrences and *Presence-Absence* surveys) and various **Environmental Predictors** data and spans 38 European countries and eight bio-geographic regions, e.g., Alpine, Atlantic, and Boreal (see Figure 2) For each species observation, we provide: (i) diverse environmental rasters (e.g., elevation, human footprint, land use, and soil), (ii) Sentinel-2-based RGB and Near-Infra-Red satellite images (128×128) with 10 m resolution, (iii) a 20-year time series of climatic variables, and (iv) satellite time-series point values for six satellite bands (R, G, B, NIR, SWIR1, and SWIR2) from the Landsat program. The data is highly diverse. Therefore, we provide a detailed description of each predictor below.

### 3.1  Species Observation Data

The species observation data comprises approximately 5 million **Presence-Only** (PO) occurrences and around 90 thousand **Presence-Absence** (PA) survey records. The PO data is the most commonly and widely available type of data and covers most European countries, but it has been sampled without any protocol, leading to various sampling biases, and the local observation of a species provides no information on the absence of others. A reporter (i.e., citizen scientist) might not have reported some species due to seasonal visibility, misidentification, or lack of interest. For both PO and PA data, we provide a short description below.

**Presence-Absence (PA) surveys.**  A presence-absence survey obtained by experienced botanists who report, as exhaustively as possible, the plant species in a given small spatial plot (usually 10–400 square meters). Hence, all species not observed during a PA survey are likely truly absent from the plot. The provided data originates from 29 source datasets[1] hosted in the European Vegetation Archive (EVA), with different spatial extents and targeted habitats. Despite the relatively large size of the PA dataset (93,703 surveys), it only covers 5,016 species—approximately half of the European flora. Besides, the distribution of these species is highly imbalanced, with most species only being observed once or twice among all the PA surveys. While constructing the training and test splits (95/5), we used a spatial block hold-out procedure [51] using a spatial grid with 10×10km cells (see the spatial grid in Figure 2) and ended up with 88,987 surveys for training and 4,716 for testing. The 10×10km test blocks were randomly selected to ensure balance in biogeographical regions.

---

[1] The extraction details and all providers are available on EVA: `https://doi.org/10.58060/qe37-tk48`

Table 1: **Presence-Only dataset sources**. Selected GBIF datasets cover 38 European countries. "Uniq. species" indicates the number of unique species in each dataset compared to the rest.

| GBIF Dataset Name | Records | Species | Uniq. species |
|---|---|---|---|
| **(our)** Pl@ntNet Observations + Pl@ntNet Occurrences | 2,298,884 | 4,631 | 295 |
| Danmarks Miljøportals Naturdatabase | 691,313 | 1,457 | 14 |
| iNaturalist Research-grade Observations | 625,681 | 7,496 | 1,754 |
| Norwegian Species Observation Service | 601,101 | 2,243 | 167 |
| Observation.org, Nature data from around the World | 241,205 | 5,108 | 429 |
| Non-native plant occurrences in Flanders and the Brussels | 178,544 | 1,464 | 134 |
| Artportalen (Swedish Species Observation System) | 163,513 | 2,771 | 464 |
| National Plant Monitoring Scheme U.K. | 120,413 | 1,109 | 11 |
| Vascular plant records verified via iRecord | 103,213 | 2,179 | 99 |
| Swiss National Databank of Vascular Plants | 49,173 | 58 | 2 |
| Invazivke - Invasive Alien Species in Slovenia | 4,171 | 60 | 1 |
| Masaryk University - Herbarium BRNU | 2,586 | 1,321 | 122 |
| GeoPlant Presence-Only data (*Combined*) | 5,079,797 | 9,709 | —— |

**Presence-Only (PO) occurrences.** A Presence-Only (PO) record is a geolocated species observation whose sampling protocol is unknown and which doesn't inform about the absence of other species. The sampling effort is highly heterogeneous in space, time, and across species. As most PO records originate from citizen-science platforms, they are generally concentrated in populated and easily accessible areas and focus on charismatic and easy-to-identify plant species. Despite this, PO data can help compensate for gaps in PA surveys when controling for sampling biases in model calibration [24, 43]. The PO data comprises 5 million records of 9,709 plant species reported between 2017 and 2021. This data originates from 13 pre-selected datasets extracted from the Global Biodiversity Information Facility (GBIF) [28], listed in and referenced in Table 1.

## 3.2 Environmental Predictor Data

The spatialized geographic and environmental predictor data are crucial for precise predictive modeling. In light of that, we have developed the biggest publicly available dataset regarding available resources and their diversity. Each survey or species observation (PO and PA) is accompanied with: (i) A four-band 128×128 satellite image at 10 m resolution around the occurrence location. (ii) Time series of the past values for six satellite bands at the point location. (iii) Various environmental rasters at the European scale (e.g., climatic, soil, elevation, land use, and human footprint variables), and (iv) Monthly time series of four climatic variables for any observation, as we provide monthly climatic rasters from 2000 to 2019. As the dataset originates from various sources and requires significant preprocessing, we thoroughly describe the acquisition process and data description below.

### 3.2.1 Environmental Rasters

We associated species observations with diverse environmental rasters, e.g., bioclimatic, soil, elevation, land cover, and human footprint. Environmental rasters used include 19 low-resolution bioclimatic rasters for Europe, nine low-resolution soil rasters describing soil properties, a high-resolution elevation raster, a medium-resolution multi-band land cover raster, and 16 low-resolution human footprint rasters (14 detailed pressures for two time periods and two summary rasters). All the environmental rasters are provided as .TIF files, reprojected to the WGS84 coordinate system (EPSG:4326) and with the same spatial extent[2], including all the species observation data.

**Land cover.** Land cover variables helped explain species distributions at all scales and significantly improved bioclimatic model performance at thinner spatial resolutions starting from 20 km [40]. Furthermore, the interactions between climate change and land cover change remain poorly understood and could strongly modify both land cover change and the distribution of threats [41]. Following that, we provide a medium-resolution multi-band land cover raster covering Europe. It is provided as a compressed GeoTIFF file with a resolution of 500m. We used the NASA earthdata portal to extract the 24 HDF raster tiles from the MODIS Terra+Aqua [26]). These HDF files stack 13 layers

---

[2]The extend is from $(min_x, min_y) = (-32.26, 26.63)$ to $(max_x, max_y) = (35.58, 72.18)$, ±1 degree.

corresponding to various land cover classifications or encoding the class confidence detailed are provided in User Guide. We merged all the HDF files into a single multi-band GeoTIFF, reprojected it to WGS84, and cropped it to the extent of GeoPlant. Each band in the provided GeoTIFF describes the land cover class prediction or its confidence under various classifications. We recommend using layers of IGBP (17 classes) or LCCS (43 classes), which are often used.

**Human footprint.** Human impact on biodiversity loss has been widely studied, resulting in 1 million species at risk of extinction. Human pressure significantly influences species extinction risk and is a better predictor of species' geographic range than biological traits [15, 16]. Therefore, we provide summary rasters combining all human pressures and detailed rasters per pressure, preserving the original data integrity. These include (i) GeoTIFF with 16 low-resolution rasters for human footprint and (ii) GeoTIFF with 14 detailed rasters across seven environmental pressures (e.g., nightlight level, population density) for two time periods (1993–2009). Both rasters go with a 1 km resolution We used global terrestrial human footprint rasters from Venter et al. [58] as reference data on human settlement and activities. Derived from remote sensing and surveys, these rasters measure direct and indirect human pressures across eight variables at a 1 km scale: built environment, population density, electrical infrastructure, cropland, pastureland, roads, railways, and navigable waterways. Except for roads and railways, each variable is available and consistent for two years: 1993 and 2009 [58]. To ensure equal pressure representation, cumulative scores are normalized by biome, per [53]. Rasters were reprojected from Mollweide to WGS84 and consistently cropped.

**Elevation.** Topography significantly affects plant species distribution by influencing light, moisture, and nutrient conditions. Including topography as a covariate in species distribution models (SDMs) has improved their performance significantly [54]. Since large-scale data on edaphic factors are still scarce, topography serves as an excellent proxy. Therefore, we provide Elevation for all available records in the form of a GeoTIFF file and a CSV file as scalar values. The raster was extracted from the ASTER Global Digital Elevation Model V3 using NASA earthdata portal.

**Soilgrids.** Physico-chemical soil properties (e.g., Ph, granulometry) are crucial to a plant species' survival ability. SoilGrids [50] is a system for global digital soil mapping that uses ML methods to map the spatial distribution of soil properties across the globe. SoilGrids prediction models are fitted at 250m resolution using over 230k soil profile observations from the WoSIS database and a series of environmental covariates. We integrated nine soil rasters corresponding to a depth of 5 to 15cm at a resolution of 30 arcsec (~1 km), i.e., the aggregated version of SoilGrids 2.0 rasters derived by resampling at 1km the mean initial predictions at 250m for each property. Nine pedologic low-resolution rasters were downloaded from ISRIC (in WGS84) and cropped to the same extent.

### 3.3 Satellite Images

Remote sensing data is a rich and globally consistent predictor variable describing the surrounding environment [31] in high resolution. Therefore we provide Sentinel-2-based RGB and Near-Infra-Red (NIR) satellite images (128×128) with 10m resolution centered on the geolocated spot and taken in the same year (see Figure 3). All images were extracted from the pre-processed rasters (composites of monthly rasters), with eliminated cloud coverage and shadows, available on the Ecodatacube. The values in extracted image patches are first thresholded at 10,000, rescaled to [0,1], and a gamma correction of 2.5 is applied (i.e., values are powered by 1/2.5). Finally, the values are rescaled to [0,255] and rounded for uint8 encoding. This process avoids using a range for high reflectance values (>10,000 in uint16) and gives more range to values close to zero, which are the most common.

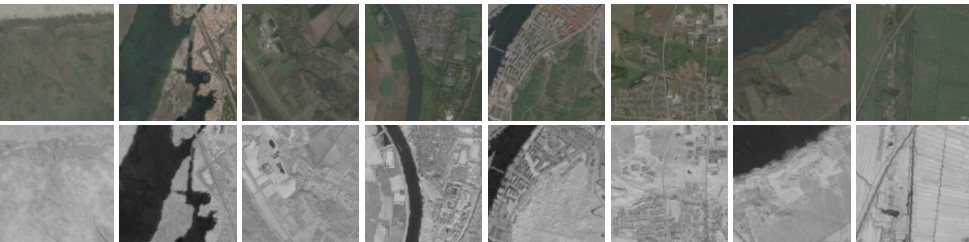

Figure 3: **Satellite image data**. 128×128 images from Sentinel-2. First row RGB, Second row NIR.

### 3.3.1 Satelite Time-series

In addition to satellite images, we provide comprehensive satellite time series data that spans over 20 years of satellite imagery. The data, obtained through the Landsat ARD program and pre-processed by EcoDataCube, covers a wide temporal range of quarterly values from 1999 to 2020, providing a detailed understanding of environmental changes over the past two decades. Each PO and PA location is linked to the time series of median point values over each season since winter 1999 for six satellite bands (R, G, B, NIR, SWIR1, and SWIR2), capturing high-resolution local signatures of seasonal vegetation changes, extreme natural events like fires, and land use changes. Due to the large size of the original rasters, data points from each spectral band were extracted for all PA and PO locations and aggregated into CSV files. A CSV file with 84 columns (representing 84 seasons from winter 1999 to autumn 2020) was created for each band. These CSV files were then aggregated into 3d tensors (cubes) with axes as [BAND, QUARTER, YEAR]. See the visualization in Figure 4.

### 3.3.2 Climatic Variables

Previous GeoLifeCLEF editions have demonstrated that climatic conditions are vital for predicting plant and animal species [36, 37]. Hence, we provide Monthly and Average Climatic rasters available at CHELSA [34]. The Monthly Climatic rasters contain four climatic variables (mean, min, and max temperature, and total precipitation) from Jan. 2000 to Dec. 2019 (960 rasters with a pixel resolution of 30 arcsecs – 1 km). The Average Climatic rasters combine 19 rasters with various averaged variables calculated from 1981 to 2010, e.g., mean annual temperature, seasonality, and extreme or limiting environmental conditions. As for the satellite time series, we pre-extracted the scalar values for all the PO and PA records and provided them as CSV files, and we aggregated them into 3d tensors with axis [RASTER-TYPE, YEAR, MONTH]. See the cube visualization in Figure 4.

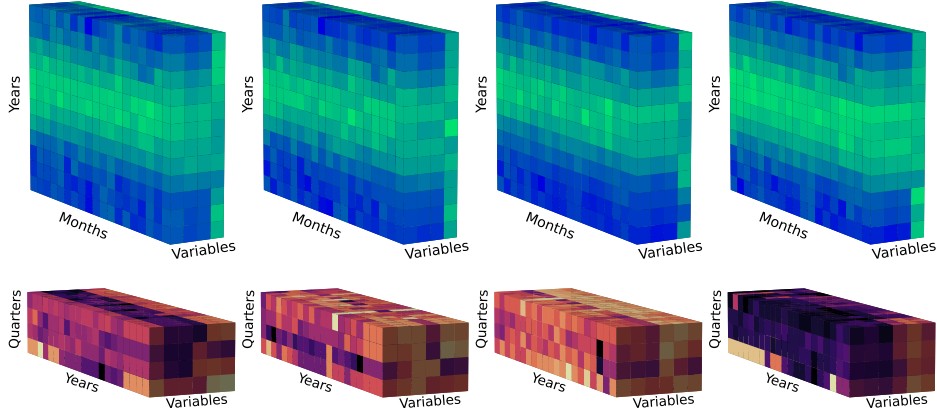

Figure 4: **Time-series data cube samples**. (Top row) – 19 years of four monthly climate variables (min + max + mean temperature, and precipitation). (Bottom row), 21 years of quarterly satellite values (R, G, B, NIR, SWIR1, and SWIR2). Each column corresponds to one PA survey. The values correspond to the pixel at the observation coordinate.

## 4 GeoPlant Benchmark

Following our positive experience, we use Kaggle to host the GeoPlant benchmark. The main benefits of the platform include (i) easy dataset use and referencing, (ii) model sharing, (iii) code development via Jupyter with free GPU resources, and (iv) straightforward setup and community management. As a part of the benchmark, we also provide/link a variety of valuable assets for deep SDMs:

- *Malpolon* [35]: A Pytorch-based framework designed for deep SDM training using various input covariates, such as bioclimatic rasters, remote sensing images, and land-use rasters.
- *Data loaders*: To allow easy loading of the raw data (for example, in test time), we provide data loaders that support loading of raw values from all GeoPlant rasters.
- *Baseline notebooks*: Jupyter Notebooks with the implementation of single and multimodal baselines. All are available in a form for direct use on Kaggle.

**Evaluation criteria.** As the provided test set is based exclusively on multi-label data from the exhaustive Presence-Absence surveys, we calculate, apart from standard AUC, the sample-averaged $F_1$-score ($F_1^s$) and Recall as a supplement. The $F_1^s$ is an average measure of overlap between the predicted and actual set of species present at a given location and time. Thus, for each test PA survey $i$ associated with a set of ground-truth labels $Y_i$ (i.e., the set of plant species reported by experts on a small grid), and provided list of predicted labels $\widehat{Y}_{i,1}, \widehat{Y}_{i,2}, \ldots, \widehat{Y}_{i,R_i}$, the $F_1^s$ is then computed as

$$F_1^s = \frac{1}{N} \sum_{i=1}^{N} \frac{TP_i}{TP_i + (FP_i + FN_i)/2},$$

$$\text{where} \begin{cases} TP_i = \text{Number of predicted species truly present, i.e. } |\widehat{Y}_i \cap Y_i|. \\ FP_i = \text{Number of species predicted but absent, i.e. } |\widehat{Y}_i \setminus Y_i|. \\ FN_i = \text{Number of species not predicted but present, i.e. } |Y_i \setminus \widehat{Y}_i|. \end{cases}$$

## 5  Some Weak and Strong Baselines

In this section, we briefly describe the various baselines trained on Presence-Absence (PA) data based on conventional methods (e.g., XGBoost) and deep neural networks (e.g., MLP and ResNets). Additional experiments with Presence-Only (PO) are available in Appendix B For all the experiments, we share fully reproducible code through GitHub. For most runs, logs are also stored on Weights and Biasses. Details about used hyperparameters, optimization strategy, etc., supporting reproducibility of all achieved and reported results, are further elaborated in Appendix C.

**Naive baselines.** With the dense and numerous observation data, one can naively predict the species' presence just by selecting a set of the most common species within administrative or bio-geographical regions. In our initial experiments (reported in Appendix B), we show that selecting top-25 most common species from PA metadata based on district & bio-geographical zone resulted in a $F_1^s$ of 20.6%. Using the same approach but with the PO data resulted in an $F_1^s$ below 9%.

**Single modality baselines with PA data.** We evaluate three architectures and modalities over the PA observations to demonstrate the potential of different modalities and the importance of multimodal approaches. We take ResNet-18 [30] (previously used in SDM) as a baseline and compare it to two custom and lighter architectures (a smaller, ResNet-6-like model and a simple MLP) on all modalities separately with five different seed values. Following the *naive baseline*, we predict only top-25 species, i.e., those with the highest logit. Results are listed in Table 2.

Table 2: **Performance of selected architectures with Presence-Absence data; single modality**. The ResNet-6 provides competitive performance to ResNet-18 in terms of all metrics and modalities and with a fraction of the computational cost. MLP badly underperformed. $F_1^s$, and Recall calculated over top-25 predictions. Values were averaged over five random seeds.

| Architecture | Climatic cubes | | | Landsat cubes | | | Sentinel-2 images | | |
|---|---|---|---|---|---|---|---|---|---|
| | AUC | Recall | $F_1^s$ | AUC | Recall | $F_1^s$ | AUC | Recall | $F_1^s$ |
| MLP | $82.8_{\pm 0.7}$ | $32.1_{\pm 0.7}$ | $22.2_{\pm 0.4}$ | $82.6_{\pm 0.1}$ | $42.0_{\pm 0.2}$ | $28.4_{\pm 0.1}$ | $71.8_{\pm 0.5}$ | $23.2_{\pm 0.7}$ | $15.8_{\pm 0.3}$ |
| ResNet-18 | $90.5_{\pm 0.2}$ | $\mathbf{37.8_{\pm 0.5}}$ | $26.2_{\pm 0.3}$ | $91.8_{\pm 0.3}$ | $44.2_{\pm 0.3}$ | $29.9_{\pm 0.2}$ | $\mathbf{88.6_{\pm 0.3}}$ | $\mathbf{33.2_{\pm 0.6}}$ | $\mathbf{22.7_{\pm 0.2}}$ |
| ResNet-6 | $\mathbf{91.8_{\pm 0.3}}$ | $37.5_{\pm 0.3}$ | $\mathbf{26.2_{\pm 0.2}}$ | $\mathbf{92.1_{\pm 0.2}}$ | $\mathbf{44.8_{\pm 0.3}}$ | $\mathbf{30.3_{\pm 0.1}}$ | $87.3_{\pm 0.3}$ | $32.1_{\pm 0.7}$ | $22.0_{\pm 0.5}$ |

**Conventional baselines.** To evaluate how traditional approaches for species distribution modeling perform on GeoPlant, we run experiments with popular methods, e.g., XGBoost and MaxEnt. We predicted only around 500 of the most common species with both methods, as the increasing number of species led to decreased performance. For XGBoost, we also run an ablation study about the influence of each predictor on the species distribution modeling (see Table 3). The MaxEnt generally underperformed heavily (see B), achieving $F_1^s$ only around 0.18. On the other hand, XGboost achieved a competitive performance in deep models trained on a single modality but still underperformed compared to multimodal ensembles. Both approaches used the default configuration and used up to 4 predictors, e.g., climatic, location, soilgrid, and land cover variables.

Table 3: **Ablation study on XGBoost performance with a selected combination of predictors.** Overall, the most impactfull predictors are related to *location*, followed by *climatic* variables. The best performance was achieved by combining all predictors. Mixing strong (e.g., *location*) and weak predictors can reduce performance.

| | | | | | | | | | | | | |
|---|---|---|---|---|---|---|---|---|---|---|---|---|
| *Location* | ✓ | – | – | – | ✓ | ✓ | ✓ | – | – | ✓ | ✓ | ✓ |
| *Climatic* | – | ✓ | – | – | ✓ | – | – | ✓ | ✓ | ✓ | ✓ | ✓ |
| *Soilgrids* | – | – | ✓ | – | – | ✓ | – | ✓ | – | ✓ | – | ✓ |
| *Land cover* | – | – | – | ✓ | – | – | ✓ | – | ✓ | – | ✓ | ✓ |
| AUC | **89.8** | 88.9 | 82.3 | 70.9 | **90.2** | 89.5 | 89.6 | 89.3 | 89.3 | 90.2 | 90.3 | **90.4** |
| Recall | **47.6** | 46.1 | 34.2 | 25.4 | **48.7** | 45.7 | 46.5 | 45.7 | 46.1 | 48.4 | **49.0** | 48.8 |
| $F_1^s$ | **28.2** | 26.7 | 21.0 | 15.3 | **28.5** | 27.4 | 27.8 | 26.9 | 27.1 | 28.6 | **28.8** | 28.7 |

**Estimating the number of species to predict per survey.** Following upon the naive baselines, we have developed a straightforward approach for multi-label classification. Instead of finding a threshold to select present species, we add a separate regression step that estimates the number of species to predict per survey. Following [55] we do not train the model on a regression task. The output space (i.e., the number of species) is divided into 15 bins containing the same number of surveys within the training set (i.e., quantiles). The average number of species per survey within each bin is then computed. The output of our model thus becomes the expected number of species for the most likely species. In addition, the $F_1^s$ measure is not symmetrical, leading to a preference for the overestimation of $K$. For this reason, we predicted for each survey the $K$ most likely species where $K$ is the number of species estimated plus an offset, which we empirically set to five:

$$\hat{\Gamma}(\mathcal{A}) = \{\sigma_{\mathcal{A}}(k) : k \in \{1, \dots, \hat{\eta}(\mathcal{A}) + \text{offset}\}\}$$

$$\hat{\eta}(\mathcal{A}) \approx |S_n \cap \mathcal{A}|,$$

where $\mathcal{A}$ is the survey field, $S_n$ the test set presences, $|S_n \cap \mathcal{A}|$ the set of present species in the survey. Hence, $\hat{\eta}(x)$ approximates the number of species present in the survey. $\sigma_x(k)$ is $k^{\text{th}}$ species in decreasing order of probability. This resulted in the best performances (bottom row of Table 4).

**Multimodal ensemble (MME) baselines.** Our baseline multimodal models have two or three branches, each encoding a different modality, e.g., (i) Sentinel2 RGB+NIR images, (ii) climatic time series encoded in the form of data cubes (tensors) with 3 dimensions: year, month and variable type (e.g., precipitation and mean, min, and max month temperature), and (iii) Landsat remote sensing time series also encoded as data cubes (tensors) with 3 dimensions: year, quarter and frequency band (e.g., R, G, B, NIR, SWIR1, and SWIR2). All modalities are independently encoded by a small residual convolution neural network with six residual blocks (i.e., *ResNet-6* in 2). Those two or three encoders produce independent embeddings (feature vectors), which are concatenated. The resulting multimodal embedding is then passed to the last layer to perform a logistic regression for each target species (logits are computed through a fully connected linear layer and sigmoïd function). Both MME models achieved better performance compared to all other baselines; for more details, refer to Table 2 and Table 4. The diagram of a three-modality MME is provided in Figure 5.

Table 4: **Performance of multimodal ensemble (MME) with PA data and multiple modalities**. Bot MME models achieved a considerable performance improvement compared to the single modality models; in terms of all measured metrics. Besides, the proposed approach for estimating the number of species at a given location improved the $F_1^s$ performance in both cases. As expected, by including fewer predictions, Recall was reduced. However, Precision increased. $F_1^s$, and Recall calculated over top-25 predictions. Values were averaged over five random seeds.

| | Climatic + Landsat | | | Climatic + Landsat + Sentinel-2 | | |
|---|---|---|---|---|---|---|
| | AUC | Recall | $F_1^s$ | AUC | Recall | $F_1^s$ |
| MME | $93.6_{\pm 0.1}$ | $\mathbf{49.3}_{\pm 0.1}$ | $33.8_{\pm 0.0}$ | $94.0_{\pm 0.1}$ | $\mathbf{49.7}_{\pm 0.1}$ | $34.1_{\pm 0.0}$ |
| MME + *Top-K estimation* | $93.6_{\pm 0.1}$ | $45.0_{\pm 0.1}$ | $\mathbf{35.9}_{\pm 0.0}$ | $94.0_{\pm 0.1}$ | $45.3_{\pm 0.1}$ | $\mathbf{36.2}_{\pm 0.0}$ |

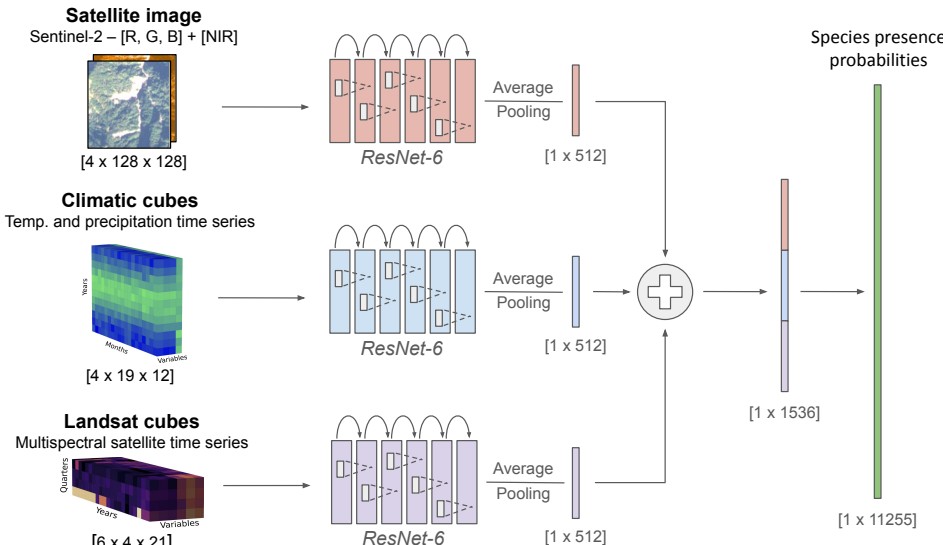

Figure 5: **Multiodal ensemble (MME) baseline**. Each modality (e.g., satellite images, climatic cube, and Landsat cube) is processed through a lightweight 6-layer residual encoder (i.e., ResNet-6). The resulting embeddings are then concatenated and passed to a final classification layer and sigmoïd.

## 6   Conclusion

This paper presents the GeoPlant dataset, a unique compilation of species observation and environmental predictor data spanning 38 European countries. With its diverse range of predictors, including satellite imagery, climatic variables, and detailed environmental rasters, the dataset provides a basis for advancing large-scale species distribution modeling (SDM). The GeoPlant dataset aims to address previous SDM datasets' limitations by offering: (i) a significantly larger and more diverse set of species occurrences, 5 million PO opportunistic observations, and 90 thousand PA survey records, which also allow for a meaningful evaluation of the SDM results, and (ii) a rich set of predictors, including not only traditional environmental factors, such as climatic variables, land cover and human footprint indices, but also medium-resolution satellite imagery and time series.

In addition to the dataset, a benchmark (hosted via a dedicated Kaggle competition) and a set of strong baselines are provided. All are easily accessible and publicly available through the Kaggle and GitHub repositories. Furthermore, all baselines are available on Kaggle in the form of Jupyter Notebooks that allow direct reproducibility for all provided baselines.

By providing an open and extensive benchmark for species distribution modeling, we hope to encourage the development of innovative modeling techniques and contribute to the broader understanding of species distribution patterns at a continental scale.

**Limitations:** Despite the significant value of the GeoPlant dataset, several limitations should be noted. First, the citizen-sourced Presence-Only (PO) data are biased toward accessible areas and common and widespread species, which may impact model accuracy. Additionally, Presence-Absence (PA) data also have limited geographic and species coverage, potentially constraining model generalizability on the European scale. Integrating heterogeneous data types – spanning diverse spatial and temporal resolutions – poses another challenge, especially on this continental scale. The dataset also reflects a significant species imbalance, with many species underrepresented, which could affect predictive performance. Finally, the multimodal nature of the dataset requires considerable computational resources for model training.

## Acknowledgements

The research described in this paper was partly funded by the European Commission via the GUARDEN and MAMBO projects, which have received funding from the European Union's Horizon Europe research and innovation program under grant agreements 101060693 and 101060639. Further models developed from this dataset will directly meet the needs of the European biodiversity strategy for 2030 through those projects. They will be used in particular to map biodiversity indicators at the European scale (e.g., presence of endangered species, invasive species, and habitat condition metrics). The authors are grateful to the OPAL infrastructure from Université Côte d'Azur for providing resources and support. Our major thanks go to thousands of European vegetation scientists of several generations who collected the original vegetation-plot data in the field and made their data available to others and those who spent myriad hours digitizing data and managing the databases in the EVA. Vegetation-plot data for this study were provided by Aaron Pérez-Haase, Adrian Indreica, Aleksander Marinšek, Alessandro Chiarucci, Ali Kavgacı, Alicia Acosta, Andraž Carni, Angela Stanisci, Anna Kuzemko, Anni Kanerva Jašková, Ariel Bergamini, Behlül Güler, Borja Jiménez-Alfaro, Corrado Marcenò, Denys Vynokurov, Emiliano Agrillo, Emin Uğurlu, Emmanuel Garbolino, Erwin Bergmeier, Eszter Ruprecht, Federico Fernández-González, Filip Küzmič, Flavia Landucci, Florian Jansen, Friedemann Goral, Gianmaria Bonari, Gianpietro Giusso del Galdo, Idoia Biurrun, Igor Lavrinenko, Ioannis Tsiripidis, Irina Tatarenko, Iris de Ronde, Iva Apostolova, Jan Jansen, Jan-Bernard Bouzillé, Jean-Claude Gégout, Jesper Erenskjold Moeslund, Joachim Schrautzer, John Janssen, John S. Rodwell, Jonathan Lenoir, Juan Antonio Campos, János Csiky, Jürgen Dengler, Kiril Vassilev, Larisa Khanina, Laura Casella, Maike Isermann, Maria Pilar Rodríguez-Rojo, Michael Glaser, Michele De Sanctis, Milan Chytrý, Milan Valachovič, Milica Stanišić-Vujačić, Mirjana Krstivojević Cuk, Olga Demina, Olivier Argagnon, Panayotis Dimopoulos, Pavel Novák, Pavel Shirokikh, Remigiusz Pielech, Renata Cušterevska, Ricarda Pätsch , Risto Virtanen, Roberto Venanzoni, Robin Pakeman, Rosario G Gavilán, Ruslan Tsvirko, Ruth Mitchell, Solvita Rūsina, Sophie Vermeersch, Stephan Hennekens, Svetlana Aćić, Svitlana Yemelianova, Sylvain Abdulhak, Tetiana Dziuba, Thomas Wohlgemuth, Tomáš Peterka, Urban Šilc, Ute Jandt, Vadim Prokhorov, Valentin Golub, Valerius Rašomavičius, Veronika Kalníková, Vitaliy Kolomiychuk, Vladimir Onipchenko, Wolfgang Willner, Xavier Font, Zygmunt Kącki, Úna FitzPatrick, and Željko Škvorc.

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

## A    Additional online resources

The complexity and size of the dataset require hosting the dataset in multiple places.

- *Self-hosted SeaFile* instance provides all resources, e.g., metadata, rasters, pre-extracted scalar values, and time-series cubes.
- Since the *Kaggle dataset* is allowing to store datasets just up to around 140Gb of size we provide all the resources but just for the Presence-Absence data.

## B    Additional Baseline Experiments

**How many species should you predict?**    We asked ourselves the same question and did a simple test where we tested the performance of a few naive baselines for each $k \in \{5, 10, 15, ..., 40\}$. From this initial experiment, it seems that the optimal $k$ value is 30; however, based on the following experiments conducted on the Landsat cubes, we discovered that the optima is actually a little bit lower, e.g., 20 and 25 for Landsat and Bioclimatic cubes respectively. For more details, see Table 5.

Table 5: **Ablation on Top-$k$ species selection.** In the Naive baseline setting, we test how the selection of the most common species affects the performance. Sample averaged $F_1^s$ score. While a higher $k$ is better for the naive approach, selecting $k = 25$ yields the best results for the most informative predictors, i.e., Landsat and Climatic cubes.

| Top-k in | Top5 | Top10 | Top15 | Top20 | Top25 | Top30 | Top35 | Top40 |
|---|---|---|---|---|---|---|---|---|
| *All Presence-Absence surveys* | 5.85 | 8.75 | 10.09 | 11.41 | 11.73 | 11.82 | 12.15 | **12.25** |
| District | 13.08 | 17.28 | 18.99 | 20.08 | 20.26 | **20.35** | 20.32 | 20.10 |
| District & Bio-geographical zone | 13.32 | 17.50 | 19.25 | 20.41 | 20.52 | **20.56** | 20.54 | 20.33 |
| ResNet-6 with Landsat cubes | 22.48 | 28.06 | 29.89 | **30.27** | 30.08 | 29.38 | 28.60 | 27.77 |
| ResNet-6 with Bioclimatic cubes | 17.36 | 22.99 | 25.34 | 26.49 | **26.98** | 26.92 | 26.51 | 25.98 |

**How far can you get with just PO data?**    The Presence-Only (PO) data consists of opportunistically collected, geolocated species observations. These observations are highly heterogeneous regarding spatial, temporal, and species coverage. Despite the availability of millions of records, which are predominantly from densely populated and easily accessible areas, PO data does not provide information on the absence of species that were not observed. This limitation poses a challenge for species distribution modeling. Our experiments confirm this issue, as illustrated in Table 6. The results fall significantly below the naive baselines, with a sample-averaged $F_1$ score ($F_1^s$) of only 20.6% when using presence-absence (PA) data. Conversely, training on PO data and testing on PA data remains a challenging problem, which is beneficial for benchmarking purposes.

Table 6: **Performance of selected architectures with Presence-Only (PO) data; single modality**. With PO data, all architectures underperformed badly compared to models trained exclusively on the PA data. $F_1^s$, Precision, and Recall calculated over top-25 predictions. Single seeds.

| | Climatic cubes | | | | Landsat cubes | | | |
|---|---|---|---|---|---|---|---|---|
| *Architecture* | AUC | Precision | Recall | $F_1^s$ | AUC | Precision | Recall | $F_1^s$ |
| MLP | 72.94 | 5.71 | 8.42 | 6.40 | 77.43 | 12.19 | 20.65 | 13.97 |
| ResNet-18 | 78.96 | **7.94** | **12.3** | **8.88** | **83.21** | **13.17** | 22.08 | **15.06** |
| ResNet-6 | 59.41 | 6.42 | 9.39 | 7.13 | 81.14 | 12.80 | **22.41** | 14.82 |

**MaxEnt baseline:**    Since the performance decreased with the increasing number of unique species, we fitted the Maxent model to predict just 492 species using four predictors, e.g., climatic variables, land cover, soilgrids, and location-related variables. For each species, the model was trained on all presence surveys (or a random sample of up to 5,000 if more existed) along with 5,000 randomly selected background surveys. The official Java implementation of Maxent was accessed through

the biomod2 R package (v4.1.2), specifically using the MAXENT.Phillips method. The Maxent model for each species is optimized as a probability distribution over sites, employing a softmax of a linear combination of expanded input covariates and L1-penalized cross-entropy minimization [47]. Therefore, Maxent outputs a score $f_i$ for each survey $i$, which is not a probability. To derive probabilities, a scaling factor $\gamma$ was optimized post-hoc by minimizing the binary cross-entropy loss across a random subset of 10,000 training surveys. The 25 species with the highest predicted probabilities were used to create the predicted species set. Two runs were evaluated: the first run achieved a test $F_1^s$ score of 0.168, using all presence data available, and the second, which balanced the training set across regions by limiting Denmark and Netherlands surveys to 5,000 each, achieved an F1 score of 0.178. The Maxent model for each species is optimized as a probability distribution over sites, employing a softmax of a linear combination of expanded input covariates and L1-penalized cross-entropy minimization [47]. Further details are available in the biomod2 documentation.

## C  Training Strategy and Hyperparameters for Baseline Experiments

For most of the PA experiments, we run a small grid search to find *optimal* values of *learning_rate*, *positive_weigh_factor*, and batch_size to the given modality and architecture. The complete set of evaluated values is available on Weights and Biasses. Here, we list only the best settings.

**Training strategy**: We split the development data into training and validation subsets, reserving a small portion (5%) for validation. While training, we use early stopping based on validation loss to prevent overfitting, storing the model that achieves the best validation $F_1^s$ score. All the models were optimized using AdamW and an adaptive learning rate scheduling, which decreases the LR by 10% each time validation loss is not decreased from one epoch to another.

**Test time**: For testing, the best-performing model (saved during early stopping) is evaluated on the test set, measuring AUC, F1, precision, and recall, which we log to Weights and Biases for a complete performance overview. All experiment configurations, training metrics, and results are also logged using Weights and Biases to allow full reproducibility.

### PA experiment – Climatic Cubes (C):
- **MLP**: batch_size = 32, learning_rate = 0.00001, num_epochs = 50, optimizer = *AdamW*, positive_weigh_factor = 10.0, random_seeds = [1, 66, 123, 777, 999]

- **ResNet-18**: batch_size = 32, learning_rate = 0.00001, num_epochs = 50, optimizer = *AdamW*, positive_weigh_factor = 1.0, random_seeds = [1, 66, 123, 777, 999]

- **ResNet-6**: batch_size = 32, learning_rate = 0.0001, num_epochs = 50, optimizer = *AdamW*, positive_weigh_factor = 10.0, random_seeds = [1, 66, 123, 777, 999]

### PA experiment – Landsat Cubes (L):
- **MLP**: batch_size = 32, learning_rate = 0.00001, num_epochs = 50, optimizer = *AdamW*, positive_weigh_factor = 1.0, random_seeds = [1, 66, 123, 777, 999]

- **ResNet-18**: batch_size = 32, learning_rate = 0.001, num_epochs = 50, optimizer = *AdamW*, positive_weigh_factor = 1.0, random_seeds = [1, 66, 123, 777, 999]

- **ResNet-6**: batch_size = 32, learning_rate = 0.0001, num_epochs = 50, optimizer = *AdamW*, positive_weigh_factor = 1.0, random_seeds = [1, 66, 123, 777, 999]

### PA experiment – Satellite Images (I):
- **MLP**: batch_size = 32, learning_rate = 0.0001, num_epochs = 50, optimizer = *AdamW*, positive_weigh_factor = 1.0, random_seeds = [1, 66, 123, 777, 999]

- **ResNet-18**: batch_size = 32, learning_rate = 0.0001, num_epochs = 50, optimizer = *AdamW*, positive_weigh_factor = 1.0, random_seeds = [1, 66, 123, 777, 999]

- **ResNet-6**: batch_size = 32, learning_rate = 0.0001, num_epochs = 50, optimizer = *AdamW*, positive_weigh_factor = 1.0, random_seeds = [1, 66, 123, 777, 999]

**PA experiment – Multimodal**

- **L + C**: batch_size = 32, learning_rate = 0.00001, num_epochs = 50, optimizer = *AdamW*, positive_weigh_factor = 1.0, random_seeds = [1, 66, 123, 777, 999]

- **L + C + I**: batch_size = 32, learning_rate = 0.00001, num_epochs = 50, optimizer = *AdamW*, positive_weigh_factor = 1.0, random_seeds = [1, 66, 123, 777, 999]

**All PO experiments:**

- **MLP**: batch_size = 4096, learning_rate = 0.0001, num_epochs = 50, optimizer = *AdamW*, positive_weigh_factor = 10.0, scheduler = *Cosine Annealing*, random_seed = 69

- **ResNet-18**: batch_size = 4096, learning_rate = 0.0001, num_epochs = 50, optimizer = *AdamW*, positive_weigh_factor = 10.0, scheduler = *Cosine Annealing*, random_seed = 69

- **ResNet-6**: batch_size = 4096, learning_rate = 0.001, num_epochs = 50, optimizer = *AdamW*, positive_weigh_factor = 10.0, scheduler = *Cosine Annealing*, random_seed = 69


# Datasheet

## Motivation

**For what purpose was the dataset created?**
We provide a user-friendly harmonized dataset and a standardized, transparent evaluation scheme to assist ecological modelers and machine learning practitioners in predicting species communities across different areas. This forms a crucial benchmark for the field.

**Who created this dataset and on behalf of which entity?**
The dataset was created by Lukas Picek[1], Christophe Botella[1], Maximilien Servajean[2], César Leblanc[1], Rémi Palard[1], Théo Larcher[1], Benjamin Deneu[1], Diego Marcos[1,3], Pierre Bonnet[4] and Alexis Joly[1] who are affiliated with [1] INRIA, [2] Université Paul Valéry, [3] Université de Montpellier, and [4] CIRAD – UMR AMAP.

**Who funded the creation of the dataset?**
The dataset was mainly funded by the Horizon Europe projects MAMBO (grant 101060639) and GUARDEN (grant 101060693).

**Any other comments?**
None.

## Composition

**What do the instances that comprise the dataset represent?**
Each "datapoint" of the dataset represents the observation of one or several plant species at a given place and time associated with environmental data describing this locality. There are two distinct types of instances: (i) the Presence-Absence (PA) surveys, which exhaustively report the plant species that are present, and (ii) the Presence-Only (PO) records, which only report one of the present species.

**How many instances are there in total?**
In total, there are 93,703 PA surveys and 5,079,797 PO records.

**Does the dataset contain all possible instances or is it a sample?**
It is, indeed, a sample of globally existing PA and PO data.

**What data does each instance consist of?**
Each instance contains the list of species observed, reduced to one species for PO instances, along with the geolocation, its spatial uncertainty, the day and year of the observation, and the various environmental data that describes the location at that time: (i) a four-band [R, G, B, NIR] satellite image at 10m resolution around the location, (ii) a multi-band time series of satellite values over the past 20 years preceding the observation at that place, (iii) the value of various other provided environmental variables at that location (e.g., climatic variables, soil physicochemical properties).

**Is there a label or target associated with each instance?**
The PA data is associated with multiple labels, i.e., a list of observed species. The PO data with just one.

**Is any information missing from individual instances?**
By definition, some species are not reported for the PO instances, which is part of the difficulty of the targeted problem. Besides, some environmental variables, like along coastlines, are missing for a minor part of the instances.

**Are relationships between individual instances made explicit?**
Yes, in a relational way through dedicated identifiers.

**Are there recommended data splits?**
A train-test split has already been proposed based on a partitioning of the data relevant to evaluating species predictions in a balanced way across European regions and habitats. The splitting procedure is described, allowing users to use a similar splitting procedure among the training data.

**Are there any errors, sources of noise, or redundancies in the dataset?**
The majority of the data come from a citizen-science platform. Therefore, there may be errors in the name of the reported species of an instance, in its geolocation, or in the satellite or environmental values associated with it, as the latter arise from diverse, complex processing schemes.

**Is the dataset self-contained, or does it link to external resources?**
It is self-contained. Links to external data sources are provided if relevant.

**Does the dataset contain data that might be considered confidential?**
No.

**Does the dataset contain data that might be offensive or cause anxiety?**
No.

**Does the dataset relate to people?**
No, the identity of people who contributed to the data collection is not provided in any way, but the identity of the institutions that assembled and processed the component data sources is provided and credited.

**Does the dataset identify any subpopulations?**
This is irrelevant as the data describes plant species.

**Is it possible to identify individuals from the dataset?**
No.

**Does the dataset contain data that might be considered sensitive?**
The dataset comprises geolocations of certain rare and/or threatened species, but all species names were anonymized to prevent users from recovering species names from our dataset.

**Any other comments?**
None.

## Collection Process

**How was the data associated with each instance acquired?**
Based on its geolocation and time.

**What mechanisms or procedures were used to collect the data?**
The component datasets were extracted from their respective websites, such as the Global Biodiversity Information Facility (GBIF) for the PO instances, the European Vegetation Archive (EVA) for PA instances, or the EcoDataCube platform for satellite data.

**If the dataset is a sample from a larger set, what was the sampling strategy?**
Yes, the PO and PA instances were subsampled from much larger sets hosted in their respective

hosting platform (GBIF and EVA) based on diverse criteria described in our methodology and, for instance, to ensure a good degree of precision regarding the location or a temporal match with the environmental data.

**Who was involved in the data collection process and how were they compensated?**
The authors extracted and processed the data during their paid working hours.

**Over what timeframe was the data collected?**
The data collation process began in October 2023 until March 2024.

**Were any ethical review processes conducted?**
No.

**Does the dataset relate to people?**
No.

**Did you collect the data from the individuals in question directly, or obtain it via third parties or other sources?**
No. Irrelevant.

**Were the individuals in question notified about the data collection?**
No. Irrelevant.

**Did the individuals in question consent to the collection and use of their data?**
No. Irrelevant.

**If consent was obtained, were the consenting individuals provided with a mechanism to revoke their consent in the future or for certain uses?** Irrelevant.

**Has an analysis of the potential impact of the dataset and its use on data subjects**
No.

**Any other comments?**
None.

## Preprocessing/cleaning/labeling

**Was any preprocessing/cleaning/labeling of the data done?**
The preprocessing and cleaning steps are complex and described in the paper and Kaggle.

**Was the "raw" data saved in addition to the preprocessed/cleaned/labeled data?**
Yes. Anyway, due to the large size of the raw satellite data (e.g., 10m/30m resolution rasters at EU scale), we do not provide it to the "end user". The raw species observation data isn't provided either due to protect the species anonymization, but the DOI of the original PO extraction is provided.

**Is the software used to preprocess/clean/label the instances available?**
Yes. The code is available through Kaggle and GitHub.

**Any other comments?**
None.

## Uses

**Has the dataset been used for any tasks already?**
It was used for the GeoLifeCLEF 2024 model evaluation campaign that just ended.

**Is there a repository that links to any or all papers or systems that use the dataset?**
No.

**What (other) tasks could the dataset be used for?**
The dataset is primarily intended for species distribution modeling. However, as it includes multimodal data, it might also be used for research in multi-model learning.

**Is there anything about the composition of the dataset or the way it was collected and preprocessed/cleaned/labeled that might impact future uses?**
Not to our best knowledge.

**Are there tasks for which the dataset should not be used?**
No.

**Any other comments?**
None.

## Distribution

**Will the dataset be distributed to third parties outside of the entity?**
The primary intention behind the publication of this dataset is to make it publicly available.

**How will the dataset be distributed?**
The dataset is distributed through multiple channels. Kaggle, Project GitHub, and SeaFile.

**When will the dataset be distributed?**
The dataset is already available, with updates planned on a monthly basis.

**Will the dataset be distributed under a copyright or other intellectual property (IP) license and/or under applicable terms of use (ToU)?**
Yes, the dataset is published under the GPL license.

**Have any third parties imposed IP-based or other restrictions on the data associated with the instances?**
No.

**Do any export controls or other regulatory restrictions apply to the dataset or to individual instances?**
No.

**Any other comments?**
None.

## Maintenance

**Who is supporting/hosting/maintaining the dataset?**
The authors of the paper will maintain the dataset and provide additional support. The dataset is permanently hosted on Seafile and Kaggle.

**How can the owner/curator/manager of the dataset be contacted?**
Owners and dataset maintainers can be contacted via email and Kaggle forum. All email addresses are provided on Kaggle and GitHub.

**Is there an erratum?**
No.

**Will the dataset be updated?**
Yes, the dataset is planned to be updated on a monthly basis.

**If the dataset relates to people, are there applicable limits on the retention of the data associated with the instances?**
Irrelevant.

**Will older versions of the dataset continue to be supported/hosted/maintained?**
New dataset versions will most likely include bug fixes, etc. The older versions of the dataset will be hosted and available but should not be used.

**If others want to extend/augment/build on/contribute to the dataset, is there a mechanism for them to do so?**
Anyone can propose a *Merge Request*, *Feature Request*, or *Bug Report* through the Kaggle forum or GitHub.

**Any other comments?**
None.

