# OpenReview forum: "GeoPlant: Spatial Plant Species Prediction Dataset"
_NeurIPS.cc/2024/Datasets_and_Benchmarks_Track — NeurIPS 2024 Track Datasets and Benchmarks Spotlight_

### Official Review · Reviewer_YHJc · 2024-07-13
**GeoLife**

**Rating:** 8
**Confidence:** 3

**Review:**

The quality of the dataset is high, as it includes a vast amount of diverse data and is designed to address the challenges of integrating heterogeneous data. The inclusion of both PO and PA data provides a robust benchmark for evaluating SDMs.

The paper is well-written and clearly explains the dataset, its components, and its potential applications. The authors provide detailed descriptions of the data sources, sampling protocols, and the methods used to construct the dataset.

The dataset is original in its scale and diversity, providing a unique resource for species distribution modeling. The integration of multi-modal data and the inclusion of both PO and PA data make it a significant contribution to the field.

The dataset is significant for its potential to advance ecological knowledge and conservation efforts. It provides a robust tool for understanding biodiversity responses and aiding decision-making in the face of global changes.

Pros
    - Scale and Diversity: The dataset covers a large number of species and includes diverse environmental rasters, satellite images, and climatic time-series data.
    - Robust Benchmark: The inclusion of both PO and PA data provides a robust benchmark for evaluating SDMs.
    - Accessibility: The dataset is openly accessible, making it easy for researchers to start using it immediately.

Cons
    - Data Imbalance: The PA data is highly imbalanced, with most species observed only once or twice.
    - Sampling Biases: The PO data is biased towards areas with high observer density and common species.

**Strengths:**

The dataset is significant for its potential to advance ecological knowledge and conservation efforts.
The dataset is highly relevant to the broader research community, providing a robust tool for understanding biodiversity responses and aiding decision-making.
The research is of high quality, as it addresses the challenges of integrating heterogeneous data and provides a robust benchmark for evaluating SDMs.

**Additional Feedback:**

The dataset is a significant contribution to the field of species distribution modeling, and its accessibility and robustness make it a valuable resource for researchers. However, addressing the limitations will further enhance its impact and utility.

**Clarity:**

The paper is well-written and clearly explains the dataset, its components, and its potential applications.

**Correctness:**

The claims made in the submission are correct, as the dataset is constructed in a sound way and provides a robust benchmark for evaluating SDMs. The evaluation methods and experiment design are appropriate and performed correctly.

**Documentation:**

Documentation is sufficient, but Kaggle notebooks are not available (links are private).

**Ethics:**

No  ethical concerns.

**Limitations:**

The PA data is highly imbalanced, with most species observed only once or twice.
The PO data is biased towards areas with high observer density and common species.
The dataset may raise ethical concerns related to data privacy, responsible use, and legal compliance.

**Opportunities For Improvement:**

The authors could improve the dataset by addressing the imbalance in the PA data, potentially through more targeted sampling or data augmentation techniques.
The authors could reduce the sampling biases in the PO data by incorporating more diverse sampling protocols or by using techniques to control for biases in model calibration.

**Relation To Prior Work:**

The paper clearly discusses how this work differs from previous contributions, highlighting the unique features of the dataset. The references are relevant and provide a comprehensive overview of the literature.

**Summary And Contributions:**

The submission titled "GeoLife: Spacial Plant Species Prediction Dataset" presents a comprehensive dataset for Species Distribution Models (SDMs) at high spatial resolution (10-50 meters). The dataset includes over 5 million Presence-Only (PO) records and 90,000 Presence-Absence (PA) surveys, covering more than 10,000 species. It integrates diverse environmental rasters, satellite images, and climatic time-series data, making it the largest and most diverse dataset for species distribution modeling. The dataset is designed to address the challenges of integrating heterogeneous data and to provide a robust benchmark for evaluating SDMs. The authors also provide pre-trained models and baseline methods, making it accessible for further research and benchmarking.

---

> ### Author Rebuttal · Authors · 2024-08-16
>
> Thank you for the constructive feedback and for providing helpful comments to improve the paper. Please see our reply related to your feedback below.
>
> > **C1: Data Imbalance: The PA data is highly imbalanced, with most species observed only once or twice.**
>
> We agree with the statements. However, even with the opportunistic nature of the data collection, it is still hard to “record” rare or even red-listed species. We were considering removing such data, but with that, it would most likely result in an unnatural distribution; which in SDM is not wanted. Another important fact is that PA data are extremely hard and time-consuming to get. We work closely with ecologists, and we plan to increase the number of PA data considerably. However, focusing just on gathering more PA observations of rare species would bias the spatial representation of the rare species in the dataset, which is not desired as well.
>
> > **C2: Sampling Biases: The PO data is biased towards areas with high observer density and common species. The authors could improve the dataset by addressing the imbalance in the PA data, potentially through more targeted sampling or data augmentation techniques. The authors could reduce the sampling biases in the PO data by incorporating more diverse sampling protocols or by using techniques to control for biases in model calibration.**
>
> We agree with you that the PO data has “some” serious sampling biases. However, this dataset was also proposed as an evaluation ground for developing methods to deal with this problem. So, from our point of view, it is more of a feature than a bug. However, it seems that we didn't clarify that clearly in the paper. If desired, we can include a better motivation.
>
> > **C3: Documentation is sufficient, but Kaggle notebooks are not available (links are private).**
>
> That's a good hint. It looks like there are some small issues with the notebook publishing. We have moved the notebooks and tested the access on multiple devices. Can you please verify that it is accessible at the moment? You should find it in the “Code” section on [Kaggle](https://kaggle.com/datasets/133b9f3f5150623216725562539b35ab0117733d7b32e38bde89bd6501129adf).

---

> > ### Comment · Reviewer_YHJc · 2024-08-17
> >
> > It seems that only the 'Data Card' section is available at this time.

---

> > > ### Author Response · Authors · 2024-08-17
> > > **Kaggle Dataset Made Public**
> > >
> > > Thank you for letting me know! Since it seems buggy, I made the dataset publicly available.
> > > Here is the [link](https://www.kaggle.com/datasets/picekl/geoplant/code).

---

### Official Review · Reviewer_QH2a · 2024-07-24
**A significant dataset for species distribution modelling.**

**Rating:** 6
**Confidence:** 2

**Review:**

This paper represent a very significant work with an extensive dataset with multiple environmental indicators that are would be relevant for SDM and cover a wide variety of fauna species in Europe and may serve as a strong foundation to existing SDM approaches.
This dataset described in the paper would of significant interest to researchers looking to profile the distribution of fauna using remote sensing methods.

A diagram of the custom CNN used for the evaluation would have been helpful.  I note that the accuracies are quite low, some error analysis (i/e confusion matrices, ROC curve, Grad-CAM activation maps) would help researchers better understand the challenge with the dataset and explore ways to improve current machine learning methods to be more performant with the dataset. I understand that the dataset having 9000+ classes would make the error analysis unwieldly, but perhaps the authors could consider evaluating the errors at the family or genus level.

I may have missed it, but the focus on the spatial predictors without considering the temporal dynamics feels like a missed opportunity. It would be interesting to see if a 2D+1D (temporal dimension) CNN/ LSTM would improve on the classification.

**Strengths:**

The dataset is extensive with multiple environmental indicators that are would be relevant for SDM and cover a wide variety of fauna species in Europe and may serve as a strong foundation to existing SDM approaches.
This dataset would be useful to researchers looking to profile the distribution of fauna using remote sensing methods.

**Additional Feedback:**

A diagram of the custom CNN would have been helpful. Based on the description am i right to assume that the CNN similar what some would call a ResNet-6?
Can the authors share some error analysis (i/e confusion matrix, ROC curves) given that even the top-40 classification performance of the CNNs are poor. (I am going to assume that the performance of VITs is even lower given that the shallower CNNs perform better than the deeper CNNs)
I also note that different tree species often displace other tree species, does the evaluation consider the different snapshot of the distribution of species in time?

**Clarity:**

The paper is well written, however i would have preferred if the authors included a diagram of the "custom" CNN used in the benchmark.

**Correctness:**

The benchmark metric and setup are standard for classifications problems. However, the authors might want to consider including newer CNN architectures i/e efficientNet-V2, for the evaluation. The dataset is constructed in a sound manner (with the caveat of the limitations of presence-absence and presence-only surveys)

**Documentation:**

There is sufficient detail on the data collection, availability and use of the dataset. Moreover the benchmarks are available through Kaggle for ease of reproduciblity.

**Ethics:**

No, I do not expect any ethical concern with the submission

**Limitations:**

I do not foresee potential negative societal impact of the work related to this dataset.

**Opportunities For Improvement:**

A diagram of the custom CNN used for the evaluation would have been helpful.  I note that the accuracies are quite low, some error analysis (i/e confusion matrices, ROC curve, Grad-CAM activation maps) would help researchers better understand the challenge with the dataset and explore ways to improve current machine learning methods to be more performant with the dataset. I understand that the dataset having 9000+ classes would make the error analysis unwieldly, but perhaps the authors could consider evaluating the errors at the family or genus level.

I may have missed it, but the focus on the spatial predictors without considering the temporal dynamics feels like a missed opportunity. It would be interesting to see if a 2D+1D (temporal dimension) CNN/ LSTM would improve on the classification.

**Relation To Prior Work:**

The authors compare their work to other previous approaches to SDMs as well as describe traditional and machine learning approaches to SDMs.

**Summary And Contributions:**

This paper describes GeoLife, a dataset for species distribution models (SDM) with focus on European flora. The dataset includes environmental rasters such as elevation, human footprint, and soil type, as well as Sentinel-2 RGB and NIR images captured as 20-year spatio-temporal series.

---

> ### Author Rebuttal · Authors · 2024-08-16
>
> Thank you for reviewing the paper and for providing valuable feedback. We greatly appreciate your efforts. Below, you will find our detailed responses to your comments.
>
> > **C1: A diagram of the custom CNN used for the evaluation would have been helpful.**
>
> Good idea. We will add the diagram with the detailed description to the revised version (i.e., camera ready).
>
> > **C2: Based on the description, am I right to assume that the CNN is similar to what some would call a ResNet-6?**
>
> We even internally call it ResNet-6, but as it is not exactly the same and some parts are different, we decided to change the name to CNN. Would you think it is better to change it back to ResNet-6?
>
> > **C3: I note that the accuracies are quite low, some error analysis (i/e confusion matrices, ROC curve, Grad-CAM activation maps) would help researchers better understand the challenge with the dataset and explore ways to improve current machine learning methods to be more performant with the dataset. I understand that the dataset having 9000+ classes would make the error analysis unwieldy, but perhaps the authors could consider evaluating the errors at the family or genus level.**
>
> This is also a very good idea, and we should have enough space, at least in the supplementary for some additional error analysis. We agree that a confusion matrix in this case would not be the best way, therefore we will:
> - show how the error varies across species (average F1 across species and/or average AUC across species), as a function of the number of training points.
> - provide a map showing the F1 per test spatial block (which are distributed across Europe),
> - provide a barplot of the F1 per habitat type (e.g., lvl1 EUNIS).
>
> > **C4: I may have missed it, but the focus on the spatial predictors without considering the temporal dynamics feels like a missed opportunity. It would be interesting to see if a 2D+1D (temporal dimension) CNN / LSTM would improve on the classification.**
>
> Indeed, there are many possible uses of the provided data, which is, in fact, the reason we put together this dataset. Since the main contribution of this dataset is not in the experiments section, we hope that the community will explore the full potential of our data / benchmark, although performing these studies remains out of the scope of this particular piece of work.
>
> > **C5: I also note that different tree species often displace other tree species. Does the evaluation consider different snapshots of the distribution of species in time?**
>
> The dataset focuses on spatial distributions for a quite restricted time period (i.e., 2017-2021), which can be perceived as a limit due to data availability and alignment constraints. Therefore, it's unlikely that we have repeated surveys at the very same location in different years which should display such displacement. In light of that, we do not do any time-wise evaluation. However, it is a good question we might focus on in the future.
>
> > **C6: The authors might want to consider including newer CNN architectures i/e efficientNet-V2, for the evaluation.**
>
> We have tested various architectures in our initial experiments, and we observed exactly what was pointed out in your review, i.e., the larger the complexity of the CNN / ViT is, the lower the performance. If reviewer and area chair prefer, we can include some experiments about the architecture search we performed and comparison with other architectures to the supplementary.

---

> > ### Comment · Reviewer_QH2a · 2024-08-17
> >
> > Thank you for the acknowledgment of the review. I will keep the rating as is. All the best for the future work with Geolife

---

### Official Review · Reviewer_faPk · 2024-07-24
**An excellent benchmark for species distribution modeling**

**Rating:** 10
**Confidence:** 4
**Correctness:** Dataset construction seems reasonable…
**Clarity:** The paper is clearly written.

**Review:**

This is great! The dataset is novel, addressing a real scientific gap. The inclusion of the environmental covariate rasters encourages inclusion of metadata in predictions, which is generally lacking in standard ML formulations. Furthermore, lack of 'ground truth negatives' in species survey work makes model evaluation extremely difficult; the inclusion of presence-absence survey data helps address this issue.  I expect that this will be a fundamental benchmark for species distribution modeling going forward. The careful design of the dataset also likely lends it to benchmarking satellite model performance more generally.

**Strengths:**

Discussed above.

**Additional Feedback:**

Thanks for putting this together!

**Documentation:**

The Kaggle page exists and allows easy access to the data.

**Ethics:**

No concerns.

**Limitations:**

The primary limitation in this kind of work is the difficulty of obtaining ground-truth data, which the authors have attempted to address. This is surely not perfect, but I appreciate the effort.

**Opportunities For Improvement:**

The F1 score is a bit too rigid, IMO. It depends on some threshold selection, which (if done poorly) can obscure the actual quality of the underlying model. The BirdCLEF competition has settled into using class-averaged ROC-AUC as it is a) threshold-free, b) resistant to label imbalance, and c) has a nice probabilistic interpretation. in a competition setting especially, where competitors have limited time and bandwidth for testing, moving to a threshold-free metric helps competitors focus more on core model quality, and less on threshold-selection heuristics.

**Relation To Prior Work:**

The paper has an adequate related works section.

**Summary And Contributions:**

This paper presents the GeoLife dataset for species distribution modeling, which combines satellite images with robust covariate information, along with groundtruth for species presence (and sometime presence-absence).

---

> ### Author Rebuttal · Authors · 2024-08-16
>
> Thank you for your positive and encouraging review. We are pleased that you found our dataset innovative and valuable. We appreciate your recognition of the dataset's practical value. Bellow, please find our response to your comments.
>
> > **C1: The F1 score is a bit too rigid, IMO. It depends on some threshold selection, which (if done poorly) can obscure the actual quality of the underlying model. The BirdCLEF competition has settled into using class-averaged ROC-AUC as it is a) threshold-free, b) resistant to label imbalance, and c) has a nice probabilistic interpretation. in a competition setting especially, where competitors have limited time and bandwidth for testing, moving to a threshold-free metric helps competitors focus more on core model quality and less on threshold-selection heuristics.**
>
> Fair point, the average of AUC across species is also commonly used in general SDMs evaluation, but its problem is that it doesn't evaluate the model's ability to capture overall species commonness/rarity nor to capture how many species are there in a very local assemblage, while both aspects are central to the problem of "high resolution" species community predictions. Maybe it's indeed too ambitious for a time-constrained competition, and we might want to discuss the problem decomposition into (i) predicting species relative frequency across space/environment and (ii) turning species-wise predictions into the prediction of a species assemblage, but we could also argue that the GeoPlant aims at being used on the longer run. To conclude, we believe that the combination of various metrics may maybe a better approach. Given that, we will include the AUC values in our tables.

---

> > ### Comment · Reviewer_faPk · 2024-08-21
> >
> > I acknowledge the response, and will keep the current score.
> >
> > Re: Metrics -
> > Our experience running the BirdCLEF Kaggle competitions was that using a threshold-based metric leads to competitors spending significant effort on the fine-tuning of the threshold, at the expense of working on core model quality. eg, it's very easy to lose with a better model with a bad threshold selection. Once we switched to a threshold-free metric, we started getting more generally useful contributions.
> >
> > Downstream tasks usually depend on a threshold selection, but the right threshold varies by application; some projects don't mind manual review but really need high recall, and others have minimal review budget and can't tolerate false positives.
> >
> > Finally, it can be helpful to separate the core decision/detection task from the downstream statistical modeling tasks. Understanding commonness/rarity, complexity of local assemblages, and so on sound like downstream tasks which could benefit from additional modeling effort. Instead of committing to a particular downstream problem, it may be best to focus on getting the best possible general decision/detection model, as strong core models tend to simplify downstream analysis.

---

### Official Review · Reviewer_7yVJ · 2024-07-28
**Very important and useful dataset, experiments can be strengthened**

**Rating:** 7
**Confidence:** 4
**Correctness:** So far as I can tell, yes.

**Review:**

This dataset is urgently needed in AI for biodiversity monitoring, and it poses fascinating challenges for the ML community. It is very well constructed and is scoped to be useful to be both algorithmists and ecologists. However, in its current form the experimental section of the paper is quite weak. The submission also omits the required paper checklist and does not discuss limitations. I would be willing to raise my score significantly if these issues were addressed.

**Strengths:**

This dataset is urgently needed in AI for biodiversity monitoring, and it poses fascinating challenges for the ML community. It is very well constructed and is scoped to be useful to be both algorithmists and ecologists.

**Additional Feedback:**

n/a

**Clarity:**

Yes, generally. The explanation of the CNN + Top-K approach is not very clear, and in general there is not enough description of how exactly the ML models were set up and trained in the multi-label paradigm (slightly too much was left to the supplementary materials here).

**Documentation:**

Yes.

**Ethics:**

No ethics issues.

**Limitations:**

The paper includes no section on limitations or discussion of them. The required checklist also appears to be missing. These are both serious omissions.

**Opportunities For Improvement:**

In its current form the experimental section of the paper is quite weak. Notably:
- There should be experiments showing the effects of ablation of different data modalities - are all of them actually useful? Prior work has shown, for example, that in some circumstances the remote sensing data isn't all that useful.
- Only deep-learning baselines (and a simple most-common-species baseline) are considered - the authors need to consider also e.g. MaxEnt and random forest.
- It might also be interesting to train on PO data and test on PA (since this is more trustworthy), though this is an optional addition.

**Relation To Prior Work:**

The work misses a number of recent works on deep SDMs that are relevant, notably:

"SatBird: Bird Species Distribution Modeling with Remote Sensing and Citizen Science Data" (Teng et al.)
"On the selection and effectiveness of pseudo-absences for species distribution modeling with deep learning" (Zbinden et al.)

**Summary And Contributions:**

This paper presents a large and expertly designed dataset on plant occurrences, including both presence-only and presence-absence data. The authors use this data to test the efficacy of various deep learning-based methods for learning species distribution models.

---

> ### Author Rebuttal · Authors · 2024-08-16
>
> We appreciate your time and effort invested in reviewing the manuscript and the valuable feedback provided; thank you. Please find our response to your comments below.
>
> > **C1: The work misses a number of recent works on deep SDMs that are relevant, notably: "SatBird: Bird Species Distribution Modeling with Remote Sensing and Citizen Science Data" (Teng et al.)** and **"On the selection and effectiveness of pseudo-absences for species distribution modeling with deep learning" (Zbinden et al.)**
>
> We will revise the introduction and related work section and include recent work. Both recommendations are relevant.
>
> > **C2: Only deep-learning baselines (and a simple most-common-species baseline) are considered - the authors need to consider also e.g. MaxEnt and random forest.**
>
> This is a good point! We already have some MaxEnt and XGBoost baselines, but given MaxEnt's low performance, we suspect there are some issues in our implantation. More precisely, using the tabular data and multiple modalities, we achieved F1 of around 0.25 with XGBoost but just ~0.125 with MaxEnt. Once we are sure about the correctness of our experiments, we will add them to the supplementary.
>
> > **C3: It might also be interesting to train on PO data and test on PA (since this is more trustworthy), though this is an optional addition.**
>
> Due to the limited space, we had to include the PO experiment in the supplementary material. We are aware that there was no reference to the supplementary, which might cause confusion, and we will include it in the revised version.
>
> > **C4: There should be experiments showing the effects of ablation of different data modalities - are all of them actually useful? Prior work has shown, for example, that in some circumstances, the remote sensing data isn't all that useful.**
>
> Such an experiment is available on a small scale. We refer to line 274 and Table 2. The last meta columns, C + S and C + S + I, show that all the modalities have some impact. However, the Satellite images add a relatively small improvement. Given the fact that it was not clear in the current version, we will update the caption of Table 2 and add an additional section about that experiment. Besides, we will add a table into the supplementary with full ablation considering all the modalities.
>
> > **C5: The paper includes no section on limitations or discussion of them. The required checklist also appears to be missing. These are both serious omissions.**
>
> We are sincerely sorry that we accidentally did not compile the checklist for submission. We include the missing checklist in the attached pdf. Besides, as we will have one additional page ready for the camera, we will add a section to the conclusion focused on the limitations.
>
> > **C6: The explanation of the CNN + Top-K approach is not very clear, and in general there is not enough description of how exactly the ML models were set up and trained in the multi-label paradigm (slightly too much was left to the supplementary materials here).**
>
> After reading the experiments section after a while, we agree that this might not be clear enough. Given that we will have one additional page, we can and will increase the clarity of all the experiments, especially the optimization procedure and architecture design.

---

> > ### Comment · Reviewer_7yVJ · 2024-08-20
> >
> > Thank you for this thoughtful response. In response to various points raised:
> >
> > C2: I would consider adding these in the main body of the paper, not just the appendix, since the comparisons are important.
> >
> > C3: Yes, I saw the PO experiment in appendix, which was helpful. I was suggesting training on PO and testing on PA (since the PA is more trustworthy) - did you run that experiment already? Maybe I misinterpreted, but I thought the appendix experiment was training and testing on PO.
> >
> > I will revise my score to reflect the improvements the authors have noted in the manuscript.

---

> > > ### Author Response · Authors · 2024-08-20
> > >
> > > Thank you for considering our rebuttal. Here are our comments on the two remaining points.
> > >
> > > > **C2: I would consider adding these in the main body of the paper, not just the appendix, since the comparisons are important.**
> > >
> > > Sure, we can move it there. Since we will have one additional page for the camera ready, including it in the paper's main body should be easy.
> > >
> > > > **C3: Yes, I saw the PO experiment in appendix, which was helpful. I was suggesting training on PO and testing on PA (since the PA is more trustworthy) - did you run that experiment already? Maybe I misinterpreted, but I thought the appendix experiment was training and testing on PO.**
> > >
> > > Given your feedback, we will improve the clarity related to the used test data in the camera ready. FYI: In all the experiments, we used the same PA test set defined in Section 3.1 (Lines 130-133) and visualized in Figure 2.

---

### Decision · Program_Chairs · 2024-09-26

**Decision:**

Accept (Spotlight)

**Comment:**

This submission introduces a novel European-scale dataset for Species Distribution Models (SDMs) at high spatial resolution (10-50 meters), for more than 10,000 species (i.e., most of the European flora), including both presence-only and presence-absence data. It enhances this data with ESA's Sentinel-2 data, NASA's Landsat data and time-series data of climate variables. This dataset aims to connect biodiversity monitoring with machine learning as it already provides benchmark evaluations from its active kaggle community.

This submission received four thoughtful and detailed reviews with an **average rating of 7.75**. The authors have provided a detailed rebuttal to address the questions and concerns of the reviewers, which was acknowledged by the reviewers.

Based on the reviewers' feedback, the rebuttal and the discussion, and finally, the overall rating, I recommend the paper for **acceptance**. I would like to ask the authors to work through all recommended improvements as given by the reviewers for the final version of the paper.